# Transcriptome Analysis of Leg Muscles and the Effects of *ALOX5* on Proliferation and Differentiation of Myoblasts in Haiyang Yellow Chickens

**DOI:** 10.3390/genes14061213

**Published:** 2023-06-01

**Authors:** Xumei Yin, Wenna Fang, Manman Yuan, Hao Sun, Jinyu Wang

**Affiliations:** 1College of Marine and Bioengineering, Yancheng Institute of Technology, Yancheng 224000, China; 2Luohe Academy of Agricultural Sciences, Luohe 462000, China; 3College of Animal Science and Technology, Yangzhou University, Yangzhou 225000, China

**Keywords:** Haiyang Yellow Chickens, transcriptome analysis, muscle growth and development, *ALOX5*, proliferation and differentiation

## Abstract

Skeletal muscle growth and development from embryo to adult consists of a series of carefully regulated changes in gene expression. This study aimed to identify candidate genes involved in Haiyang Yellow Chickens’ growth and to understand the regulatory role of the key gene *ALOX5* (arachidonate 5-lipoxygenase) in myoblast proliferation and differentiation. In order to search the key candidate genes in the process of muscle growth and development, RNA sequencing was used to compare the transcriptomes of chicken muscle tissues at four developmental stages and to analyze the effects of *ALOX5* gene interference and overexpression on myoblast proliferation and differentiation at the cellular level. The results showed that 5743 differentially expressed genes (DEGs) (|fold change| ≥ 2; FDR ≤ 0.05) were detected by pairwise comparison in male chickens. Functional analysis showed that the DEGs were mainly involved in the processes of cell proliferation, growth, and developmental process. Many of the DEGs, such as *MYOCD* (Myocardin), *MUSTN1* (Musculoskeletal Embryonic Nuclear Protein 1), *MYOG* (*MYOG*enin), *MYOD1* (*MYOG*enic differentiation 1), *FGF8* (fibroblast growth factor 8), *FGF9* (fibroblast growth factor 9), and *IGF-1* (insulin-like growth factor-1), were related to chicken growth and development. KEGG pathway (Kyoto Encyclopedia of Genes and Genomes pathway) analysis showed that the DEGs were significantly enriched in two pathways related to growth and development: ECM-receptor interaction (Extracellular Matrix) and MAPK signaling pathway (Mitogen-Activated Protein Kinase). With the extension of differentiation time, the expression of the *ALOX5* gene showed an increasing trend, and it was found that interference with the *ALOX5* gene could inhibit the proliferation and differentiation of myoblasts and that overexpression of the *ALOX5* gene could promote the proliferation and differentiation of myoblasts. This study identified a range of genes and several pathways that may be involved in regulating early growth, and it can provide theoretical research for understanding the regulation mechanism of muscle growth and development of Haiyang Yellow Chickens.

## 1. Introduction

The growth and development of skeletal muscle include a series of changes occurring from the embryonic stage to adulthood. In poultry farming, the incubation period is a particularly critical stage, and muscle growth and development during this period directly affects the production performance of adult animals. This subsequently influences the economic benefits of the broiler industry. Therefore, the investigation of muscle growth and development in poultry facilitates the improvement of meat production performance in animals.

Embryonic development is an important stage in animals [1,2,3]. Genetic factors and maternal nutritional environment can determine the phenotypic manifestations of the growth and development of young and adult animals [4,5,6]. In the entire growth period of poultry, the development of muscle fibers affecting the production and quality of poultry muscle is completed during the embryonic development period. The number of muscle fibers produced during this stage does not change throughout the animal’s life. After birth, muscle production can only be increased in terms of thickening and lengthening the muscle fibers. In summary, the growth and production of muscle fibers during embryonic development can significantly affect the productivity of adult animals.

Jinghai Yellow Chicken is a novel broiler breed that has been jointly cultivated by Jiangsu Jinghai Poultry Group, Yangzhou University, and Jiangsu Provincial Animal Husbandry Station. Eight generations of closed breeding were performed, under the premise of retaining the advantages of the original local chickens, to cultivate a new broiler breed with identical body shape and morphological features, excellent performance in terms of production, and stable heredity. In the present study, we selected Haiyang Yellow Chicken as the test chicken. One of the parents of this chicken was the Jinghai Yellow Chicken, which was bred in previous years. The Haiyang Yellow Chicken is known for its exquisite meat quality, high reproductive performance, and strong stress resistance, and the commercial generation of this chicken can be marketed at the age of 70 days. Furthermore, the characteristics of this chicken include excellent meat quality and strong adaptability, among others.

Skeletal muscle development was a complex and rigorous regulatory process involving the proliferation and differentiation of myoblasts and their fusion into multi-nucleated muscle fibers with contractile properties [7]. In humans, dysplasia of muscle development before birth often leads to muscle disorders such as congenital muscular dystrophy and myasthenic syndrome [8]. For animal production, muscle development directly affected meat production and commercial value [9]. Therefore, the study of skeletal muscle development has important theoretical and practical significance. A series of changes occur in the expression of regulatory genes involved in muscle growth and development from the embryonic stage to adulthood. Furthermore, these gene expression changes are associated with muscle growth, and the underlying regulatory mechanisms are particularly complex. At present, a large number of key genes have been identified to be involved in the regulation of skeletal muscle growth and development. Zhang et al. found KLF4 acted as the target gene of miR-7, which played a positive role in the proliferation and differentiation of CPMs [10]. He also found that the proliferation and differentiation of CPMs were regulated by miR-27b-3p and MSTN. MIR-27b-3p could inhibit the expression of MSTN in the proliferative CPMs and promote the proliferation of CPMs. Both miR-27b-3p and MSTN could inhibit the differentiation of CPMs [11]. Hou found that overexpression of CKM can inhibit the proliferation of primary myoblasts and promote apoptosis of primary myoblasts, while inhibition of the CKM gene could promote proliferation and inhibit apoptosis of primary myoblasts [12]. Mo et al. showed that ApoD promoted the proliferation and differentiation of chicken myoblasts by reducing intracellular ROS content, reducing the ubiquitination of MyoD by Atrogin 1, and inhibiting MyHC by MuRF1 [13].

In the present study, we aimed to identify the differentially expressed genes (DEGs) that affected muscle growth and development from the embryonic stage to adulthood and study the influence of key genes on the proliferation and differentiation ability of myoblasts. RNA-Seq technology and bioinformatics analysis were used to screen and annotate the DEGs involved in the process of muscle growth and development. We consequently noted that the key gene arachidonate 5-lipoxygenase (*ALOX5*) may affect muscle growth and development. This result was verified by constructing the siRNA interference sequence and overexpression vector of the *ALOX5* gene to study the effect of the *ALOX5* gene on the proliferation and differentiation ability of myoblasts. This was performed as a preliminary verification of the *ALOX5* gene function. This study can provide a theoretical basis for understanding the regulation mechanism underlying muscle growth and development in Haiyang Yellow Chicken.

## 2. Materials and Methods

### 2.1. Ethics Statement

All animal experiments were performed according to the protocol of the Animal Use Committee of the Chinese Ministry of Agriculture and were approved by the Animal Care Committee of the Department of Animal Science and Technology, Yangzhou University. Necessary efforts were undertaken to minimize animal suffering (202103-325).

### 2.2. Experimental Animals and Tissues

The test samples belonged to the same batch of Haiyang Yellow Chicken (commercial generation) from Jiangsu Jinghai Poultry Group Co., Ltd. (Nantong, China). The semi-sibling individuals with favorable body characteristics were selected and raised in the same environment. Three roosters were slaughtered at 12 days (early stage of skeletal muscle differentiation), 16 days (late stage of skeletal muscle differentiation), the first day after birth (early stage of skeletal muscle hypertrophy), and 10 weeks of age (age to market). Tissues were collected from the leg muscles and the fascia and cartilage were removed. These tissue samples were placed in enzyme-free frozen storages tubes, flash frozen with liquid nitrogen, and stored at −80 °C in an ultra-low temperature refrigerator until RNA extraction. The fertilized eggs of Haiyang Yellow Chicken were provided by Jiangsu Jinghai Poultry Group Co., Ltd., and incubated at 38 °C and 60% relative humidity. Primary chicken myoblasts were cultured from leg muscles at the embryonic age of 12 days.

### 2.3. RNA Isolation, Library Preparation, and Sequencing

Total RNA was isolated from each sample using the TRIzol regent (Invitrogen, Carlsbad, CA, USA). The purity, concentration, and integrity of the RNA were checked using a NanoDrop 2000 spectrophotometer (Thermo Fisher Scientific, Waltham, MA, USA) and an Agilent 2100 Bioanalyzer (Agilent Technologies, Santa Clara, CA, USA), respectively. The RNA integrity number (RIN) of all samples was >6.5, OD260/280 ≥ 1.8.

Approximately 3 µg of RNA was used per sample to construct a complementary DNA (cDNA) library, according to the following procedures: the ribosome RNA (rRNA) was removed and strand-specific RNA-Seq libraries were subsequently using rRNA-depleted RNA. After RNA fragmentation, the double-stranded cDNA was synthesized by replacing dTTPs (deoxythymidine triphosphate) with dUTPs (deoxyuridine triphosphate) in a reaction buffer used for second-strand cDNA synthesis. The resulting double-stranded cDNA was ligated to adaptors, after being end-repaired and A-tailed. Single-strand cDNA was then obtained using USER (Uracil-Specific Excision Reagent) Enzyme (NEB, Ipswich, UK). Finally, a polymerase chain reaction (PCR) was performed to enrich the cDNA libraries. Sequencing was performed on an Illumina Hiseq 2500 instrument using the TruSeq Cluster Kit v3-cBot-HS (Illumina, San Diego, CA, USA) to generate 150 bp paired-end reads.

### 2.4. Quality Control and Comparative Analysis

Raw data were subjected to quality control examinations using the Quality Control tool for High Throughput Sequence Data FastQC v0.11.2 [14]. The Phred scores (Q20, Q30) and G + C content of the raw data were analyzed. Clean data were concurrently obtained by discarding reads containing adapter or ploy-N and low-quality reads (>50% of base with a Phred score of <5) from the raw data.

### 2.5. Sequencing Data Analysis and Transcriptome Assembly

The FASTQC (http://www.bioinformatics.babraham.ac.uk/projects/fastqc/, accessed on 10 October 2022) was used to assess raw data quality. Low-quality reads (reads containing adaptors, unknown bases, and low-quality bases) were removed using NGSQC Toolkit v2.3.3. Clean reads obtained through filtering were mapped to the chicken reference genome (NCBI) using the TopHat software. The mapped reads were used for additional transcript annotation and to calculate the expression level using the FPKM (fragments per kilobase per million reads) method. The DESeq package (http://bioconductor.org/packages/release/bioc/html/DESeq.html, accessed on 21 December 2022) was used to calculate differences in gene expressions. Genes with false discovery rate (FDR) < 0.05 and |log2(FoldChange)| > 2 were considered DEGs between the two groups. These DEGs were consequently subjected to Gene Ontology (GO) annotation and enrichment analysis (NCBI) using the classic algorithm and Fisher’s exact test. Then, further characterization of the metabolic pathways of DEGs involved analyzing the pathways using the Kyoto Encyclopedia of Genes and Genomes (KEGG) database.

### 2.6. Validation of Gene Expression by Quantitative PCR Analysis

The total RNA from the 12 samples used for the RNA-Seq experiment was amplified by quantitative PCR (qPCR). Single-strand cDNA was synthesized using the PrimeScriptTM RT Master Mix kit (Vazyme Biotech Co., Ltd., Nan-jing, China). qRT-PCR was performed using an Applied Biosystems 7500 Real-Time PCR System (Life Technologies, Gaithersburg, MD, USA) with specific primers. qRT-PCR amplification was performed in 20-µL reaction volumes containing 1 µL of cDNA, 10 µL of SYBR Premix Ex Taq polymerase (2×) (Vazyme Biotech Co., Ltd., Nanjing, China), 0.4 µL of ROX Reference Dye II (50×), 0.4 µL of the forward primer (10 mmol/L), 0.4 µL of the reverse primer (10 mmol/L), and 6.8 µL of dH_2_O. The amplification started with an initial denaturation step at 95 °C for 30 s followed by 40 cycles at 95 °C for 5 s and an annealing step at 60 °C for 34 s; at this point, fluorescence was observed. Finally, a dissociation curve to test PCR specificity was generated by one cycle at 95 °C for 15 s followed by 60 °C for 1 min and ramped to 95 °C with acquired fluorescence. Specific primers were designed based on sequences retrieved from the National Center for Biotechnology Information (NCBI) database (Appendix A). Therefore, in the present study, HSP70 and β-actin genes selected as internal reference genes were calculated using the normalization factors based on the geometric means of these two reference genes quantities. The 2^−ΔΔCt^ method was used to transform the data from the relative quantification [15].

### 2.7. Isolation and Culture of Chicken Primary Myoblast (CPM)

CPM isolation and culture: primary myoblasts were isolated from the leg muscle of 12-day-old chicken embryos. First, the muscle tissues were separated from the skin and bone and then homogenized in a petri dish. To release single cells, the suspension was digested with pancreatin for 20 min at 37 °C. After neutralization with a complete medium, single cells were collected via centrifugation at 500× *g*. Subsequently, serial plating was performed to enrich primary myoblasts and eliminate fibroblasts. Primary myoblasts were cultured in DMEM-F12 (Gibco, Grand Island, New York, USA) medium with 20% FBS and 1% penicillin-streptomycin-amphotericin B Solution. All cells were cultured at 37 °C in a 5% CO_2_ humidified atmosphere.

### 2.8. RNA Oligonucleotides and Plasmids Construction

Three small interfering RNAs (siRNAs) and non-specific siRNA negative control of the *ALOX5* gene were designed and synthesized by Shanghai Gemma Gene Biotechnology Co., Ltd. (Shanghai, China); furthermore, the oligonucleotide sequences used in this study are presented in Table 1. To construct the *ALOX5* overexpression plasmid, the full-length sequence of *ALOX5* was amplified from the chicken breast muscle cDNA via PCR and cloned into the expression plasmidpcDNA-3.1 vector by using the XbaI and XhoI restriction sites.

### 2.9. Cell Transfection

When the cell density was 60–70%, the plasmids and siRNA were transfected according to the instructions of the transfection reagent. After 24–36 h, the cells were collected by adding TRIzol reagent in the enzyme-free Ep tube.

### 2.10. Cell Proliferation Capacity Assay

#### 2.10.1. EdU Assay

Myoblast cells seeded in 12-well plates were cultured to 60% density and then transfected. *ALOX5*-siRNA, *ALOX5*-NC, overexpression vector, and blank vector were transfected into CPM. After 24 h, fixation, and staining were performed according to EdU cell proliferation detection kit instructions (Ruibo Biology, Guangzhou, China). The number of stained cells was observed and counted using a fluorescent inverted microscope to compare the cell proliferation between the experimental and control groups.

#### 2.10.2. CCK-8 Assay

Myoblasts were inoculated in 96-well plates. When the cell density reached approximately 30%, the cells were transfected with *ALOX5*-siRNA, *ALOX5*-NC, overexpression vector, and blank vector, with eights replicates per group. According to the instructions for CCK-8 Cell Counting Kit (Vazyme Biotech Co., Ltd., Nanjing, China). At four time points, i.e., 12, 24, 36, and 48 h after transfection, the absorbance values were measured at 450 nm using an enzyme marker, which indirectly reflected the number of cells.

### 2.11. Cell Differentiation Ability Assay

#### 2.11.1. Immunofluorescence

Cells were seeded in 12-well plates. After transfection for 48 h, the cells were fixed in 4% formaldehyde for 30 min before washing them three times with PBS for 5 min. The cells were subsequently permeabilized by adding 0.2% Triton X-100 for 15 min and blocked with goat serum for 30 min. The cells were incubated with the primary antibody Desmin (1:400) overnight at 4 °C against light and with the CY3-labeled secondary antibody (Sheep Anti-Rabbit IgG, 1:400) at 37 °C for 1 h (against the light). The cell nuclei were stained with DAPI for 5 min. Images were obtained using a fluorescence microscope. The area of cells labeled with anti-Cy3 was examined using ImageJ software, and the total myotube area was calculated as a percentage of the total area covered by myotubes.

#### 2.11.2. Changes in the Expression of Genes Related to Cell Differentiation

The *MYOG*, MYOD, and MEF2C genes are involved in cell differentiation. A comparison of the mRNA expression levels of these genes in the experimental and control groups was performed to determine the influence of *ALOX5* gene interference or overexpression on the myoblast differentiation ability.

### 2.12. Statistical Analyses

All data are presented as mean ± standard error of the mean (S.E.M.) based on at least three independent experiments for each treatment. Unpaired Student’s *t*-test was used to calculate the *p*-values, and *p* < 0.05 indicated statistical significance.

## 3. Results

### 3.1. Transcriptome Sequencing Results of Tissue RNA Samples

The results of the detection of 12 RNA samples are presented in Appendix A. The purity and integrity of the extracted RNA conformed to the requirements of the library sequencing and may be used for the subsequent RNA-Seq experiment.

### 3.2. Comparative Analysis of Reads and Reference Genome

HISAT software was used to conduct genome localization analysis of filtered sequencing sequences, and the statistical analysis results were shown in Appendix A. It can be seen from the table that the total mapped of each sample reaches more than 84.63%, indicating that there is no contamination and alignment to the genome is correct. Multiple mapped is below 6.02%, meeting the requirement below 10%. Uniquely mapped to more than 78.61%.

### 3.3. Screening and Functional Analysis of DEGs

#### 3.3.1. Screening of DEGs

We used the DESeq software package to analyze DEGs in the leg muscles of Haiyang Yellow Chickens at different stages of growth. The screening conditions were the presence of DEGs (Fold Change ≥ 2 and FDR ≤ 0.05). Following the elimination of duplicate values, 5743 DEGs were screened in male chickens (Figure 1 and Figure 2A). Among these DEGs, 1818 were E12 vs. E16 (909 upregulated and 909 downregulated), 3494 were E12 vs. 1d (1848 upregulated and 1646 downregulated), 3777 were E12 vs. 10W (1989 upregulated and 1788 downregulated), 1248 were E16 vs. 1d (738 upregulated and 510 downregulated), 1882 were E16 vs. 10W (1026 upregulated and 856 downregulated), and 866 were 1d vs. 10W (274 upregulated and 592 downregulated). The number of upregulated DEGs was higher than that of downregulated DEGs. Further analysis of the inter-DEG interaction was performed through the construction of Venn diagrams with the DEGs of E12 vs. E16, E16 vs. 1d, 1d vs. 10W, and E12 vs. 10W in male chickens (Figure 2B). A total of 21 DEGs were concurrently expressed in the four comparison groups, among which 12 genes were annotated (Table 2), including AHSG, PTPRT, CLEC19A, *ALOX5*, ACACB, DSG2, AP1S3, CACNG5, MYBPH, PHGDH, MYH1C, and NTRK2. However, nine genes are not annotated thus far.

#### 3.3.2. GO Functional Annotation and Enrichment Analysis of DEGs

All DEGs were annotated with GO functions, which primarily included the following three parts: molecular function, biological process, and cellular component. GO enrichment function analysis was performed for 5743 DEGs (*p* < 0.05), and the results revealed 56 significantly enriched GO items, and the main enriched GO items were essentially the same, including cellular process, biological regulation, regulation of the biological process, metabolic process, binging, catalytic activity, cell, and cell components (Appendix A). Several of the abovementioned DEGs showed a remarkable degree of enrichment and association with the development of muscles (Table 3), such as cell proliferation and the growth and developmental process. Several genes associated with muscle growth and development were included, such as *MYOCD*, *MUSTN1*, *MYOG*, *MYOD1*, *FGF8*, FGF6, and *IGF-1*, among others.

#### 3.3.3. Pathway Enrichment Analysis of DEGs

The function of DEGs in different growth stages of Haiyang Yellow Chicken was elucidated through the identification of 5743 DEGs (FC ≥ 2, FDR < 0.05), and pathway enrichment analysis showed that 35 DEGs significantly enriched pathways (*p* < 0.05). The 20 most significantly enriched pathways are presented in Appendix A. Many of these pathways were involved in the biosynthesis of amino acids; valine, leucine, and isoleucine degradation; alanine, aspartate, and glutamate metabolism; phenylalanine, tyrosine, and tryptophan biosynthesis; arginine biosynthesis; propanoate metabolism; glycolysis/gluconeogenesis; carbon metabolism; and PPAR (peroxisome proliferators-activated receptors) signaling pathway. In addition, certain signaling pathways related to muscle growth and development in male chickens were identified (Table 4), e.g., the ECM-receptor interaction and MAPK signaling pathway. Although they were not included in the top 20 significant enrichment pathways, they were significant enrichment pathways in chickens, and they play important regulatory roles in the muscle growth and development of chickens.

### 3.4. qRT-PCR Validation of DEGs Obtained via RNA-Seq

We verified the accuracy of RNA-Seq results by randomly selecting six DEGs, i.e., S100A10, TMEM33, GST2L, NDUFB4, COX7B, and SLC25A5, from male chickens and subjected to qRT-PCR. The reliability of the experiment was confirmed by ensuring that the RNA samples used for qRT-PCR were identical to those used for RNA-Seq library-building sequencing. Expressions of the six target genes are presented in Appendix A. We noted that the expression trends of each gene in different growth stages of Haiyang Yellow Chickens were fundamentally consistent with the results of RNA-Seq analysis, and the correlation between RNA-Seq and qRT-PCR was <0.01, which indicated that the fluorescence quantitative results of each gene showed a significant correlation with the results of RNA-Seq, which further explained the accuracy of the sequencing results.

### 3.5. Spatial and Temporal Expression of the ALOX5 Gene during Myoblast Differentiation

The newly extracted myoblasts were inoculated on a 6-well plate, and differentiation was induced through the administration of 2–4% fetal bovine serum when the cell density reached 70% (Figure 3). Cells were collected at eight time points, i.e., at cell densities of 50% and 100% and at 1–6 days after differentiation (DM1–DM6). The TRIzol reagent was used to extract cellular RNa. β-actin and HSP70 were used as internal references for qRT-PCR to detect *ALOX5* gene expression at different time points of myoblast differentiation. Before induction of differentiation, we noted a low relative *ALOX5* gene expression in myoblasts, and this value did not change significantly when the cell density was 50% and 100% (Figure 4). Furthermore, extending the differentiation time resulted in a significant increase in the *ALOX5* gene expression, and it reached the highest level on the fifth day of differentiation. The *ALOX5* gene expression at DM3, DM4, DM5, and DM6 was significantly higher than that at 50% cell density (*p* < 0.01); this highlighted the critical role of the *ALOX5* gene in the differentiation of chicken myoblasts.

### 3.6. Effect of ALOX5 Gene on Myoblast Proliferation

To investigate the effect of the *ALOX5* gene on myoblast proliferation, EdU staining, and CCK-8 assay were performed. The experimental group was divided into four treatment groups, pcDNA3.1-*ALOX5* overexpression group, pcDNA3.1 blank control group, si-*ALOX5* interference group, and si-NC negative control group. In the EdU experiment, we stained the four experiment groups 24 h after transfection and observed them under a fluorescent inverted microscope. The results showed (Figure 5A,B and Figure 6A,B) that the cell proliferation ability of pcDNA3.1-*ALOX5* overexpression group was significantly higher than that of pcDNA3.1 blank control group (*p* < 0.01), and the cell proliferation ability of si-*ALOX5* interference group was significantly lower than that of si-NC negative control group (*p* < 0.05); In the CCK-8 experiment, the cell proliferation capacity was detected at four time points, 12 h, 24 h, 36 h and 48 h after transfection, respectively. As shown in Figure 5C and Figure 6C, except for 12 h, the cell proliferation ability of the pcDNA3.1-*ALOX5* overexpression group at 24 h, 36 h, and 48 h was significantly higher than that of the pcDNA3.1 blank control group (*p* < 0.05). The cell proliferation capacity of the si-*ALOX5* interference group was significantly lower than that of the si-NC negative control group at four time points (*p* < 0.01). In conclusion, overexpression of the *ALOX5* gene can promote the proliferation of myoblasts, while interference of the *ALOX5* gene can inhibit the proliferation of myoblasts. Therefore, we can conclude that the *ALOX5* gene can promote the proliferation of chicken myoblasts.

### 3.7. Effect of ALOX5 Gene on Myoblast Differentiation

To investigate the effect of *ALOX5* gene interference or overexpression on myoblast differentiation, we also divided the experimental group into four treatment groups, namely the pcDNA3.1-*ALOX5* overexpression group, pcDNA3.1 blank control group, si-*ALOX5* interference group, and si-NC negative control group. 48h after transfection (when muscle tubes were obviously present), staining was performed by immunofluorescence method, and the proportion of muscle tube area in the total area of each experimental group was compared. The results showed that (Figure 7A,B and Figure 8A,B) the proportion of muscle tube area in total area in the pcDNA3.1-*ALOX5* overexpression group was significantly higher than that in the pcDNA3.1 blank control group (*p* < 0.01), furthermore, the proportion of muscle tube area in total area in si-*ALOX5* interference group was significantly lower than that in si-NC negative control group (*p* < 0.01). In addition, we changed 2%–4% fetal bovine serum to induce differentiation 6–8 h after transfection and collected cells until the cells clearly differentiated into muscle ducts. RT-PCR was used to detect the relative expression levels of differentiation-related genes such as MYOD, *MYOG*, and MEF2C, as shown in Figure 7C and Figure 8C. The expression level of differentiation-related genes after overexpression of the *ALOX5* gene was significantly higher than those in the pcDNA3.1 blank control group (*p* < 0.01), furthermore, the expression of differentiation-related genes after *ALOX5* gene interference was significantly lower than that of si-NC negative control group (*p* < 0.05). Combined with the results of immunofluorescence staining and the changes in the expression levels of differentiation-related genes, we could see that overexpression of the *ALOX5* gene could promote the ability of myotube differentiation, however, interference of the *ALOX5* gene could inhibit myotube differentiation. Therefore, we conclude that the *ALOX5* gene promotes the differentiation ability of chicken myoblasts.

## 4. Discussion

RNA-Seq technology has reportedly been used to identify genes associated with the regulation of muscle growth and development thus far. Chen Ting et al. [16] conducted transcriptome analysis on the leg muscle tissue of Guangxi chicken and determined 2304 DEGs; among these DEGs, 998 genes were upregulated, and 1306 genes were downregulated. Combined GO function and KEGG pathway enrichment analyses were used to elucidate the selected DEGs, and five genes related to muscle growth and development were subsequently obtained, including MYH10, MYH15, FGF10, FGF16, and GDF8. Piórkowska et al. [17] collected breast muscle tissues from eight pullet hens—divided into two groups with four pullets in each group—according to different muscle shear forces. They subsequently performed transcriptome analysis and noted that several genes related to muscle tenderness were screened out, such as ASB2, THRSP, and PLIN1. In the present study, transcriptome sequencing was performed on the leg muscles of Haiyang Yellow Chickens at four key periods—i.e., embryonic age at 12 days, embryonic age at 16 days, one day after birth, and 10 weeks of age. This was performed to analyze changes in transcription levels of the leg muscle tissues during the growth and development of Haiyang Yellow Chickens and to screen for key genes involved in the regulation of muscle growth and development of broilers. The Venn diagrams showed that 21 DEGs were expressed in all four comparison groups, some of which were reportedly associated with muscle development. MYH1C is involved in muscle contraction [18,19] and the MYBPH gene plays an important role in skeletal muscle formation [20]. Several other genes are known to regulate cell proliferation, growth, and differentiation. The PHGDH gene can promote the proliferation and differentiation of chicken myoblasts [21]. PTPRT plays an important role in a series of important life activities such as cell growth, metabolism, migration, proliferation, differentiation, ion channel regulation, and immune response. DSG2 critically influences cell adhesion [22], epithelial cell proliferation, and tumor formation [23]. The *ALOX5* gene plays an important role in the self-renewal, proliferation, and differentiation of mouse myeloid leukemia stem cells; furthermore, it is involved in the proliferation and differentiation of myeloid leukemia cells, which is essential for the identification of the malignant biological characteristics of leukemia stem cells [24]. Based on the function of these genes, we speculated that these genes may be related to muscle growth and development and cell activity and play an important role in the regulation of muscle growth and development.

The differential expression of genes associated with growth and development is regarded as the main etiology for the occurrence of genetic variation in broilers. Among the DEGs related to muscle growth and development identified in this study, some are closely associated with muscle growth and development. Zheng et al. [25] showed that the *MUSTN1* gene promotes the growth of skeletal muscles in broilers and laying hens. Therefore, the *MUSTN1* gene may be a regulatory gene specific to the development of skeletal muscles. MYF5, MYF6, *MYOG*, and *MYOD1* are important members of the MRFs family and play important regulatory roles in the process of muscle proliferation and differentiation. The members of this gene family have a common structure, i.e., bHLH (basic helix-loop-helix mechanism), which is related to protein polymerization and can form dimers and tetramers. However, it is evident that only dimers can bind to DNA to regulate the growth of muscle cells. Additionally, the members of this family do not have an independent function, and the other is to coordinate with each other to regulate the growth and development of skeletal muscle. The insulin-like growth factor one receptor plays a crucial role in signaling cell survival and proliferation. *IGF-1* expression level is reported to increase in tumor cells, which can promote tumor occurrence through the promotion of cell proliferation or inhibition of cell apoptosis [26,27]. Fibroblast growth factors are active substances that promote the growth of fibroblasts identified in the crude extracts of the brain and the pituitary gland in 1940 and were isolated and purified for the first time in 1974 [28]. The DEGs *FGF8* and *FGF9* detected are categorized as paracrine-type and are closely related to a variety of biological functions, including cell growth and differentiation, angiogenesis, and embryonic development, among others. This observation consequently indicated that these DEGs are capable of affecting muscle growth and development.

The number of muscle fibers is reportedly determined during the embryonic period [29]. Myoblasts originate in spindle cells, migrate to the appropriate site of muscle formation, and proliferate during hyperplasia. These myoblasts then exit the cell cycle, fuse to form multinucleated myotubes, and differentiate with the onset of muscle-specific protein expression. After the actual number of muscle fibers is determined during hyperplasia, the skeletal muscle stem cells, satellite cells located under the underlying layer of the muscle fiber wiki, are activated, and fuse their nuclei with the muscle fibers, resulting in a further increase in DNA and ultimately in protein synthesis. This increase in the size of the muscle fibers after birth (hypertrophy) is responsible for the increase in muscle mass [30,31]. Herein, the results of GO functional annotation analysis demonstrated that DEGs were mainly involved in cellular processes, metabolic processes, biological regulation, cell proliferation, and muscle growth and development. Many of the DEGs influencing chicken growth and development are associated with these terms, which suggested that chicken growth is a complex process influenced by multiple genes and regulated by multiple GO terms.

KEGG enrichment pathway analysis was performed on DEGs of Haiyang Yellow Chickens, and two key pathways affecting muscle growth and development appeared to be significantly enriched. These enriched pathways were the ECM-receptor interaction and MAPK signaling pathway. The extracellular matrix (ECM) is a complex mixture of large, functional molecules that interact with cell surface components to directly or indirectly regulate cell adhesion, migration, differentiation, proliferation, and apoptosis [32]. ECM makes up the connective tissue around muscles, and it is reported to interact with growth factors, regulate cell signaling pathways, and affect the growth and development of muscle fibers [33,34,35]. MAPK signaling pathway can be activated by receptor tyrosine kinase, G protein-coupled receptors, and certain cytokine receptors, and transmit extracellular signal stimulation to cells, thereby causing cell proliferation, apoptosis, differentiation, and a series of reactions. This subsequently promotes the occurrence and development of tumors [36]. MAPK signaling pathway is primarily used in skeletal muscle growth and development [37,38]. Therefore, these two signaling pathways are believed to play important regulatory roles in the growth and development of rooster muscles in Haiyang Yellow Chickens.

The process of muscle development is complex and may be regulated by numerous genes or transcription factors throughout the process. Some regulatory genes or transcription factors that affect muscle growth and development have been reported; however, several potentially related genes or transcription factors that require further elucidation exist. The *ALOX5* gene is a common DEG used to compare different growth stages of Haiyang Yellow Chickens, and the *ALOX5* gene is reported to play an important role in the proliferation and differentiation of leukemia stem cells [24]. However, studies on the effect of the *ALOX5* gene on the proliferation and differentiation of myoblasts are lacking. Considering this data paucity, we used chicken myoblasts as experimental materials to construct the overexpression vector of the *ALOX5* gene and siRNA interference fragment, and transfected chicken myoblasts to analyze the effect of the *ALOX5* gene on the proliferation and differentiation ability of chicken myoblasts. The observations reported in the present study will lay a theoretical foundation for further investigation of the effect of the *ALOX5* gene on muscle development.

Myoblasts are precursor cells of skeletal muscle fibers. After several cycles of proliferation, the cells fuse to form muscle tubes and finally differentiate to form mature muscle fibers. Therefore, myoblast proliferation and differentiation are believed to be the primary driving force of embryonic skeletal muscle development. The differentiation process of myoblasts is extremely complex and involves the activity of numerous genes that play a regulatory role in this process. However, these genes function via different mechanisms and at different times, and some work at specific times and under specific circumstances [39]. In the present study, we studied the expression profile of the *ALOX5* gene in myoblasts during differentiation. Our results demonstrated that the expression level of the *ALOX5* gene was very low in myoblasts before differentiation and in the early stage of differentiation. Considering the progress of cell differentiation, the expression level of the *ALOX5* gene increased significantly. Therefore, it was our understanding that the *ALOX5* gene has a certain degree of influence on the differentiation ability of myoblasts. At present, reports on the effect of the *ALOX5* gene on myoblasts are lacking; however, the *ALOX5* gene has been reported to be associated with the pathogenesis of many diseases, such as asthma, allergic rhinitis and other allergic diseases, cardiovascular diseases, multiple sclerosis, and tumors [40]. The fact that *ALOX5* is not expressed in most normal epithelial cells has been reported extensively, whereas *ALOX5* and its pathway-related proteins are significantly upregulated in a plethora of tumor cells, including CRC (colorectal cancer) and esophageal cancer, and their expression is closely related to tumor metastasis and poor prognosis of patients [41]. In addition, LTB4, the metabolite of *ALOX5*, can influence ERK1/2 to promote CRC cell proliferation or enhance the secretion of TNF-α and heparin combined with epidermal growth factor-like growth factors through the upregulation of the expression of matrix metalloproteinases, thereby promoting the occurrence and development of tumors [42]. Therefore, the *ALOX5* gene plays a role in tumor cell proliferation.

Upon further examination of the effect of the *ALOX5* gene on myoblast proliferation and differentiation was achieved by designing interference fragments and overexpression vectors of the *ALOX5* gene. Furthermore, changes in cell proliferation and differentiation after *ALOX5* gene interference and overexpression were detected. The results showed that the *ALOX5* gene can promote not only the proliferation but also the differentiation of myoblasts. Currently, numerous studies have examined genes and transcription factors that regulate muscle development. Niu Pengfei [43] noted that the MYBPH gene can inhibit both the proliferation and differentiation of myoblasts. Additionally, Cai et al. [44] found that miR-16-5p can directly act on the SESN1 gene and regulated the p53 signaling pathway, thereby affecting the proliferation and apoptosis of myoblasts, and participating in the differentiation process of myoblasts. Cui et al. [45] studied the expression level of the MYCN gene at the proliferation, differentiation, embryonic, and production and development stages of muscle cells, and found that the expression level of the MYCN gene at the proliferating stage, and it gradually increased with the increase in differentiation time. However, at the growth and development stage from birth to the age of 98 days, the expression level of the MYCN gene at 0–14 days of age showed a sharp decline. After 14 days of age, the change was not significant and maintained at a low level. These results indicated that the MYCN gene is involved in the regulation of muscle tissue development, and its function is predominantly observed in the late stage of chicken embryo development to two weeks after birth. Gan et al. [46] found that Leptin could regulate the proliferation and differentiation of duck myoblasts through the activation of AMPK and P13K signaling pathways; however, the action of the AMPK signaling pathway should be greater than that of the P13K signaling pathway. Myoblast proliferation and differentiation are known to play a crucial role in the maturation of myoblasts. In the process of muscle development, myoblast differentiation is regulated by many genes. Although there have been many reports on the key genes regulating myoblast differentiation, the process of myoblast differentiation is very complex; this indicates that the scope of further research on this subject is remarkably vast. At present, numerous studies have examined the genes that are known to affect cell differentiation. For example, the MyoD gene is involved in the early stages of muscle differentiation, which can promote the differentiation of other cells into muscle cells, and it plays a decisive role in the transcription of specific muscle genes [47,48]. However, *MYOG* and Myf6 genes primarily play a role in the late stage of myocyte differentiation, which can promote the differentiation of myocytes into myotube muscle fibers [49]. They are important members of the MRFs family and have important regulatory roles in the process of muscle differentiation. In this experiment, the total area of myotube after overexpression. Furthermore, we noted that the total area of myotube after overexpression of the *ALOX5* gene accounted for a higher proportion than that in the control group, and the relative expression levels of differentiation-related genes *MYOD1*, *MYOG*, and MEF2C were significantly increased. However, the experimental results after interference with the *ALOX5* gene contradicted the other results reported herein. Therefore, our preliminary conclusion was that the *ALOX5* gene was involved in the differentiation process of myoblasts. However, the regulation mechanism warrants further study.

Myocyte differentiation and other processes would be evident only when the cell proliferation accumulated to a certain extent. The proliferation of myosatocytes was regulated by many regulatory factors alone or together. However, they play considerably different regulatory roles, some activate myosatellite cells, some promote cell proliferation, and some inhibit cell apoptosis. For example, HGF (hepatocyte growth factor) is a multifunctional growth factor that can activate satellite cells [50,51]. Pax3 (pairing box gene 3), as a myoblast enhancer, can promote cell proliferation and inhibit cell apoptosis [52]. In this study, the effect of the *ALOX5* gene on myoblast proliferation was investigated. After *ALOX5* gene interference and overexpression, EdU staining was detected at 24h after transfection, and CCK-8 assay was performed at 12, 24, 36, and 48 h after transfection, respectively. The results showed that *ALOX5* gene interference inhibited cell proliferation, and overexpression promoted cell proliferation. The results of EdU and CCK-8 experiments were consistent, which further demonstrated that the *ALOX5* gene can promote the proliferation of myoblasts.

## Figures and Tables

**Figure 1 genes-14-01213-f001:**
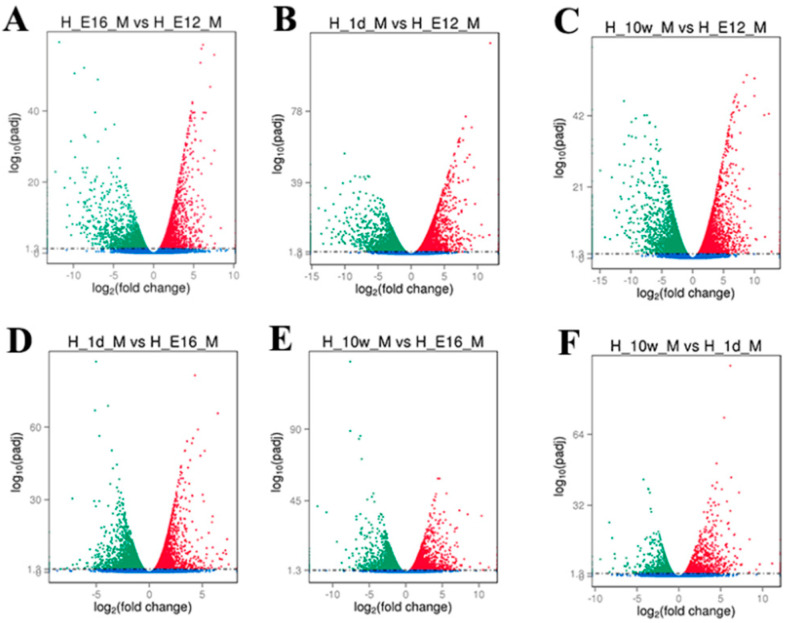
The volcano of DEGs in male chicken. (**A**) Volcano map of differentially expressed genes for E12 vs. E16; (**B**) volcano map of differentially expressed genes for E12 vs. 1d; (**C**) volcano map of differentially expressed genes for E12 vs. 10w; (**D**) volcano map of differentially expressed genes for E16 vs. 1d; (**E**) volcano map of differentially expressed genes for E16 vs. 10w; and (**F**) volcano map of differentially expressed genes for 1d vs. 10w.

**Figure 2 genes-14-01213-f002:**
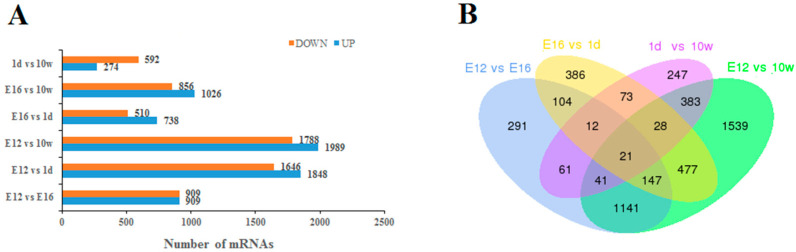
Differentially expressed genes during chicken muscle growth and development at four different stages. (**A**) Numbers of upregulated and downregulated genes in male chickens; (**B**) Venn diagram of differentially expressed genes in four comparisons of male chickens (E12-M vs. E16-M, E16-M vs. 1d-M, 1d-M vs10W-M, and E12-M vs. 10W-M).

**Figure 3 genes-14-01213-f003:**
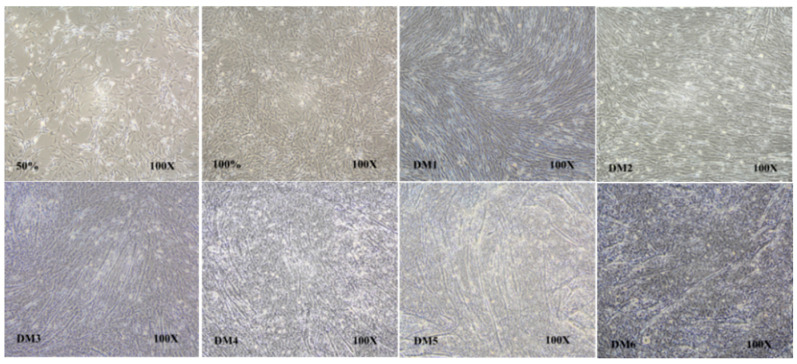
The comparison of myoblast proliferation and differentiation in different stages. Where 50% and 100%, respectively, represent the density of cells before induction differentiation. DM1–DM6 represents the fusion state of muscle tubes 1–6 days after induction of differentiation.

**Figure 4 genes-14-01213-f004:**
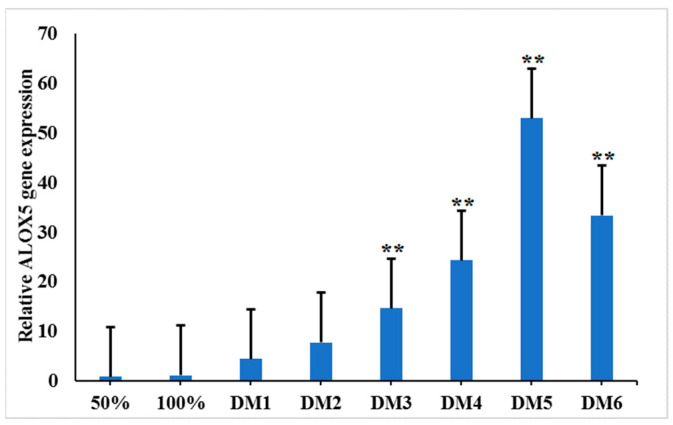
Relative expression trend of *ALOX5* gene during myoblast differentiation. ** represent very significant difference (*p* < 0.01).

**Figure 5 genes-14-01213-f005:**
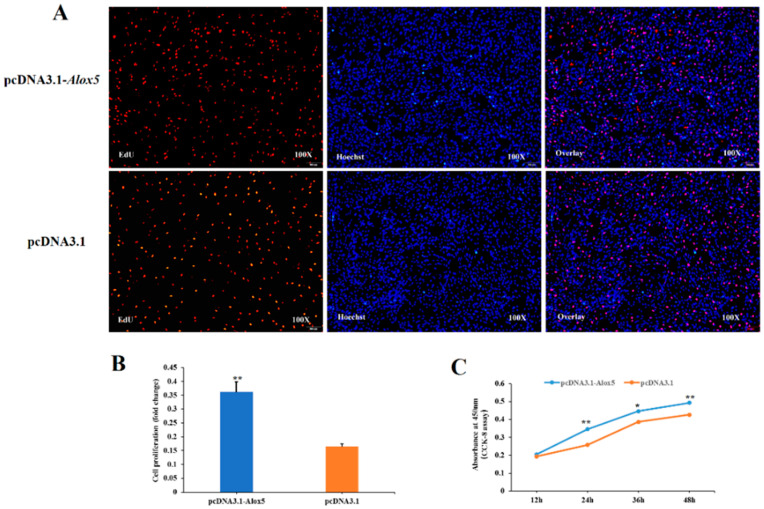
The effect of over-expression of *ALOX5* gene on the proliferation of myoblast. * represent significant difference (*p* < 0.05); ** represent very significant difference (*p* < 0.01). (**A**) The results of EdU assay (The proportion of red cells to blue cells was statistically analyzed by Image-Pro Plus software). (**B**) The numbers of proliferative cells (Microscopic results of *ALOX5* gene overexpression in myoblasts). (**C**) The results of CCK-8 cell proliferation detection assay.

**Figure 6 genes-14-01213-f006:**
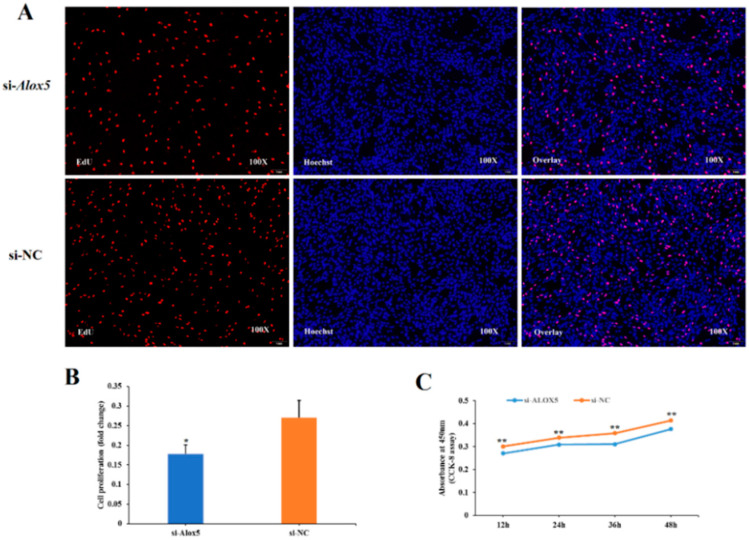
The effect of interfer *ALOX5* gene on the proliferation of myoblast. * represent significant difference (*p* < 0.05); ** represent very significant difference (*p* < 0.01). (**A**) The results of EdU assay (the proportion of red cells to blue cells was statistically analyzed by Image-Pro Plus software). (**B**) The numbers of proliferative cells (microscopic results of *ALOX5* gene interference in myoblasts). (**C**) The results of CCK-8 cell proliferation detection assay.

**Figure 7 genes-14-01213-f007:**
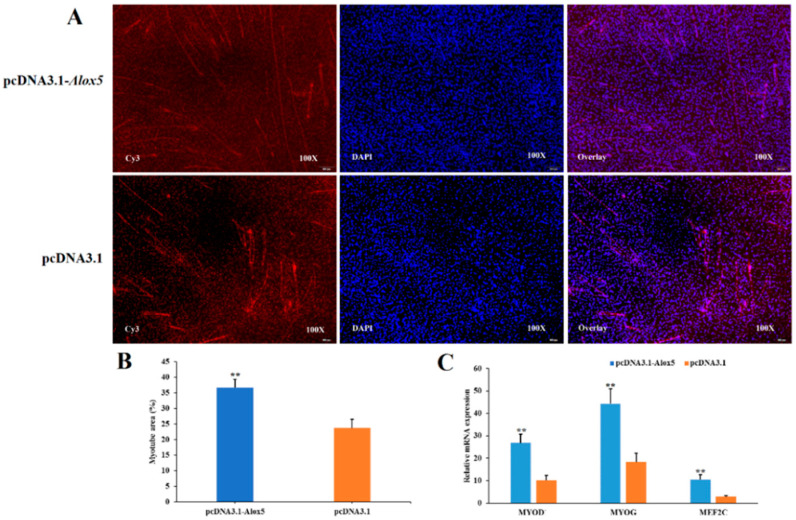
The effect of over-expression *ALOX5* gene on the differentiation of myoblast. ** represent very significant difference (*p* < 0.01). (**A**) Immunofluorescence analysis of Cy3-staining cells. (**B**) The proportion of myotube area (%). (**C**) The detection of differentiation-related gene expression.

**Figure 8 genes-14-01213-f008:**
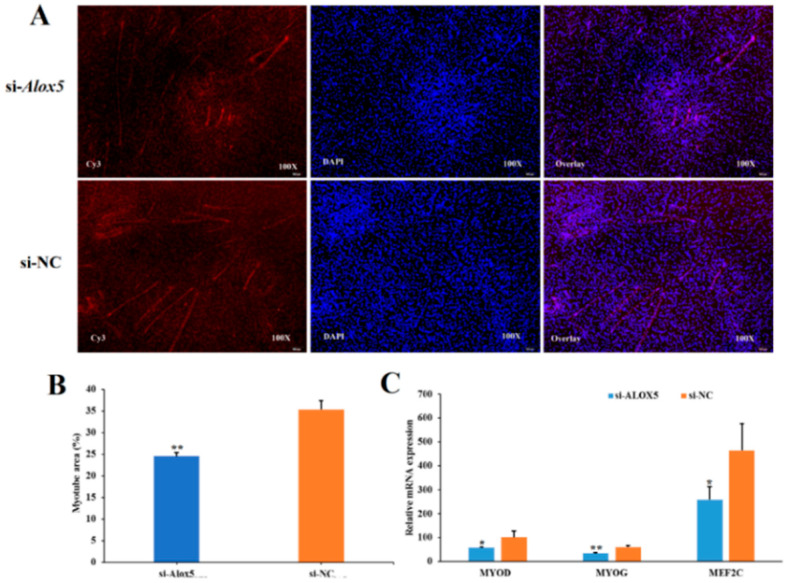
The effect of interfer *ALOX5* gene on the differentiation of myoblast. * represent significant difference (*p* < 0.05); ** represent very significant difference (*p* < 0.01). (**A**) Immunofluorescence analysis of Cy3-staining cells. (**B**) The proportion of myotube area (%). (**C**) The detection of differentiation-related gene expression.

**Table 1 genes-14-01213-t001:** The information on *ALOX5* gene interference sequence and negative control sequence.

Primer Name	Primer Sequence (5′-3′)	Sequence Size (bp)
*ALOX5*-Gallus-578	CCAGUUUGACAACGAGAAATT	21
	UUUCUCGUUGUCAAACUGGTT	21
*ALOX5*-Gallus-1000	CCAUCUGUUUGCUGUACAATT	21
	UUGUACAGCAAACAGAUGGTT	21
*ALOX5*-Gallus-1160	CCACCAGACGGUGACCCAUTT	21
	AUGGGUCACCGUCUGGUGGTT	21
*ALOX5*-Gallus-NC	UUCUCCGAACGUGUCACGUTT	21

**Table 2 genes-14-01213-t002:** Co-differentially expressed genes in male chicken.

Annotated		Description
ENSGALG00000008601	AHSG	α 2-HS glycoprotein
ENSGALG00000037889	PTPRT	protein tyrosine phosphatase, receptor type T
ENSGALG00000030546	CLEC19A	C-type lectin domain family 19 member A
ENSGALG00000005857	*ALOX5*	arachidonate 5-lipoxygenase
ENSGALG00000005043	ACACB	acetyl-CoA carboxylase β
ENSGALG00000015142	DSG2	desmoglein 2
ENSGALG00000005171	AP1S3	Gallus gallus adaptor-related protein complex 1, sigma 3 subunit (AP1S3)
ENSGALG00000003911	CACNG5	calcium voltage-gated channel auxiliary subunit γ 5
ENSGALG00000000164	MYBPH	Gallus gallus myosin binding protein H (MYBPH)
ENSGALG00000002988	PHGDH	phosphoglycerate dehydrogenase
ENSGALG00000032404	MYH1C	Gallus gallus myosin, heavy chain 1C, skeletal muscle
ENSGALG00000012594	NTRK2	Gallus gallus neurotrophic tyrosine kinase, receptor, type 2 (NTRK2)

**Table 3 genes-14-01213-t003:** The gene ontology terms related to the growth and development of male and female chickens.

GO Terms ID	Function Description	Number of DEGs	Partial Genes Associated with Muscle Growth and Development
GO:0032502	developmental process	994	MYBPHL, *MYOG*, *MYOCD*, *MUSTN1*, MYLK3, MYL2, MYL3, GDF2, GDF10, *MYOD1*, MYLK2, FGF4, *FGF8*, MYBPC3, FGFR2, MYF6, ACTA1, EGF, IGF-I, MYBPC1, MEF2C, FGF14, MYOC, *FGF9*, FGF19, IGFBP3, IGF2BP1
GO:0008283	cell proliferation	325	*MYOG*, *MYOCD*, *MUSTN1*, GDF2, *MYOD1*, FGF4, *FGF8*, FGFR2, EGF, IGF-I, MEF2C, FGFR3, GAS6, *FGF9*, FGF19, IGF2BP1
GO:0040007	growth	173	*MYOCD*, *MUSTN1*, MYL2, GDF2, *MYOD1*, *FGF8*, IGF-I, MEF2C, *FGF9*, IGFBP3, GDF8

**Table 4 genes-14-01213-t004:** Pathways involved in growth and development.

KEGG Term	Number of DEGs	*p*-Value	Partial genes Associated with Muscle Growth and Development
ECM-receptor interaction	31	0.00009	LAMB3, COMP, LAMC1, COL9A3, CD36, ITGB8, SPP1, THBS2, VTN, TNR,
MAPK signaling pathway	77	0.013	FGF4, *FGF8*, FGFR2, EGF, IGF1, FGFR3, TGFA, *FGF9*, FGF19, FGF22

## Data Availability

Not applicable.

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
