# Peer review of "Transcriptome Analysis of Leg Muscles and the Effects of ALOX5 on Proliferation and Differentiation of Myoblasts in Haiyang Yellow Chickens"

_genes, 2023, doi:10.3390/genes14061213_

Round 1

Reviewer 1 Report

Comments

Abstract is poorly written. You write Objective, Methods, Results and Conclusions. Add 2-3 lines of result part in the abstract section and how this research is useful.

Add Transcriptome analysis in the Keywords section of the manuscript.

Kindly include some latest references in the introduction section of the manuscript.

Add one more paragraph of introduction section and write about the Importance of the study and future aspect.

Line no 43-76 no reference?

In the journal Genes, After Introduction then Materials and Methods

Line no 432-436 provide the ethical No?

Line no 111 provide the details legends

In Figure 1 improve the image quality

Line no 152 provide the details legends

Line no 166 provide the details legends

Line no 183 provide the details legends

Line no 185 provide the details legends

Line no 207 provide the details legends

Line no 211 provide the details legends

Many figures are thee in the manuscript. Maximum you can give 8 figures, rest you can put it in supporting figure

Kindly check the grammatical mistake throughout the manuscript.

Minor editing of English language required

Author Response

Dear editor,

Thank you for your letter and for the reviewers’ comments concerning our manuscript. We have made correction which we hope meet with approval. Revised portion are marked in red in the paper. The main corrections in the paper and the responds to the reviewer’s questions are as flowing:

  1. Abstract is poorly written. You write Objective, Methods, Results and Conclusions. Add 2-3 lines of result part in the abstract section and how this research is useful.

Thank you for the reviewer’s suggestion. I had added Objective, Methods, Results and Conclusions to the summary, and also added a description of the results section and the usefulness of the study.

  1. Add Transcriptome analysis in the Keywords section of the manuscript.

Thanks for your suggestions. I had added transcriptome analysis to the keywords.

  1. Kindly include some latest references in the introduction section of the manuscript.

Thank you very much. Some recently references on muscle growth and development had been added to the introduction.

  1. Add one more paragraph of introduction section and write about the Importance of the study and future aspect.

Thank you for your guidance. The importance of the study and future aspect had been added to the introduction.

  1. Line no 43-76 no reference? In the journal Genes, After Introduction then Materials and Methods.

We appreciated for your careful check. We had confirmed that line 43-76 had no references, and I had moved Materials and Methods after Introduction.

  1. Line no 432-436 provide the ethical No?

We appreciated for your careful check. We had added the ethical No in the manuscript.

  1. In Figure 1 improve the image quality.

Thanks for your suggestions. I had improved the image quality.

  1. Line no 111, 152, 166, 183, 185, 207, 211 provide the details legends.

Thank you for the reviewer’s suggestion. We had provided the details legends in the manuscript.

  1. Many figures are thee in the manuscript. Maximum you can give 8 figures, rest you can put it in supporting figure.

Thanks for your suggestions. We had moved some figures to the supplemently material.

Once again, thank you very much for your guide and comments

Best Regards,

Sincerely yours

Xuemei Yin

College of Marine and Bioengineering, YanCheng Institute of Technology, Yancheng, 224100, Jiangsu, China

E-mail: xmyin1990@163.com

Reviewer 2 Report

Success of finding 12 DE genes is solely depending on RNA sequencing in this study. Authors should read the instruction manual for data deposit section. Authors should realize that all the RNA sequencing data should deposit data either in the GEO database or in the NCBI’s Sequence Read Archive (SRA) for reviewers to examine the data quality and for further usage. I highly recommend authors to do that and share their data.

Author Response

Dear editor,

Thank you for your letter and for the reviewers’ comments concerning our manuscript. We have made correction which we hope meet with approval. Revised portion are marked in red in the paper. The main corrections in the paper and the responds to the reviewer’s questions are as flowing:

  1. Success of finding 12 DE genes is solely depending on RNA sequencing in this study. Authors should read the instruction manual for data deposit section. Authors should realize that all the RNA sequencing data should deposit data either in the GEO database or in the NCBI’s Sequence Read Archive (SRA) for reviewers to examine the data quality and for further usage. I highly recommend authors to do that and share their data.

Thank you for your advice. We are uploading the data to NCBI and will add the link to the article later.

Once again, thank you very much for your guide and comments

Best Regards,

Sincerely yours

Xuemei Yin

College of Marine and Bioengineering, YanCheng Institute of Technology, Yancheng, 224100, Jiangsu, China

E-mail: xmyin1990@163.com

Reviewer 3 Report

The study describes an attempt to reveal gens and mechanisms involved in muscle development and myoblasts differentiation in selected breed of chicken. In my opinion the study is interesting and well design, however, I am only able to assess RNA-Seq and qPCR part of experiment. Another reviewer is needed to evaluate methodology of ALOX5 gene effect on myoblasts differentiation.

I am sure that manuscript would strongly benefit from revision by a native speaker. It contains several language errors that cannot be corrected within this review, especially when I am not a native speaker.

I have some comments that need to be addressed:

1.       Introduction about the muscle tissue should be more related to meat production, not with exercise or health. First two sentences of introduction could be deleted.

2.       What is “mother’s feeding environment” in birds? Please reconsider.

3.       “Embryonic development is indispensable step” – how could be different? Please rewrite.

4.       Please remove/replace awkward expressions like e.g.: “high-quality broiler”, “cultivate new quality broiler”, “self-bred” and many others.

5.       Section 2.1. includes very basic technical information on RNA quality. Please move to the supplementary material.

6.       Section 2.2. does not give any important information. Please add or replace with data such as: number of reads generated per sample, number and percentage of reads that passed filtering, number and percentage of reads that were mapped to the reference genome, number of reads mapped uniquely. Such information will allow to assess the quality of your RNA-Seq data.

7.       Because methodology is at the end of the manuscript, initial results lacs the information what are E12, 16 and remaining abbreviations. Please move the experiment design description to the begging of the results.

8.       Table 1 is poorly informative. Please add gene description, regulation across conditions etc.  Please rewrite title to be more precise.

9.       GO analysis in section 2.3.2. does not involve overrepresentation tests? Why? Please add. The Figure 3 is poorly informative – move to supplements. Please remove sentence “In addition, 125 we noted that the proportion of classified items of biological processes was greater than 126 that of molecular functions and cell components (Figure 3)”. Table 2 does not present overrepresentation tests. It is poorly informative.

10.   Figure 4 – please reconsider figure title

11.   Table 3. I understand that you are presetting p-value, not adjusted p or FDR? No significant values? Please clarify.

12.   Section 4.5. - in methods section, reference genome and annotation file accession number plus database used is enough. Please provide instead of whole link.

13.   Section 4.5. - FPKM is not recommended for further normalization with Deseq2 (I understand you used Deseq2 not old Deseq?). You should use raw counts for differential expression analysis. Please reconsider.

14.   Raw sequencing data as well as processed data e.g. raw counts should be submitted to the public database such as SRA and/or GEO.

15.   It is not clear how many replicates were prepared per condition. If less than 3, I would be forced to change my recommendation on the manuscript.

I am sure that manuscript would strongly benefit from revision by a native speaker. It contains several language errors that cannot be corrected within this review, especially when I am not a native speaker.

Author Response

Dear editor,

Thank you for your letter and for the reviewers’ comments concerning our manuscript. We have made correction which we hope meet with approval. Revised portion are marked in red in the paper. The main corrections in the paper and the responds to the reviewer’s questions are as flowing:

  1. Introduction about the muscle tissue should be more related to meat production, not with exercise or health. First two sentences of introduction could be deleted.

Thanks for your suggestions. I had deleted the first two sentences of introduction.

  1. What is “mother’s feeding environment” in birds? Please reconsider.

Thank you for the reviewer’s suggestion. I had changed “mother’s feeding environment” to “maternal nutritional environment”.

  1. “Embryonic development is indispensable step” – how could be different? Please rewrite.

Thanks for your advices. What I’m trying to say is that embryonic development is very important, and it had been revised in the manuscript.

  1. Please remove/replace awkward expressions like e.g.: “high-quality broiler”, “cultivate new quality broiler”, “self-bred” and many others.

Thank you for the reviewer’s suggestion. I had removed some awkward expressions.

  1. Section 2.1. includes very basic technical information on RNA quality. Please move to the supplementary material.

Thank you very much.

  1. Section 2.2. does not give any important information. Please add or replace with data such as: number of reads generated per sample, number and percentage of reads that passed filtering, number and percentage of reads that were mapped to the reference genome, number of reads mapped uniquely. Such information will allow to assess the quality of your RNA-Seq data.

Thank you for your guidance. We had removed non-essential information and added information to help the quality of your RNA-Seq data.

  1. Because methodology is at the end of the manuscript, initial results lacs the information what are E12, 16 and remaining abbreviations. Please move the experiment design description to the begging of the results.

Thanks for your suggestions. Materials and methods had been put into the second part as required, and experimental results into the third part.

  1. Table 1 is poorly informative. Please add gene description, regulation across conditions etc.  Please rewrite title to be more precise.

Thanks for your advices. We had added the gene description to the table.

  1. GO analysis in section 2.3.2. does not involve overrepresentation tests? Why? Please add. The Figure 3 is poorly informative – move to supplements. Please remove sentence “In addition, 125 we noted that the proportion of classified items of biological processes was greater than 126 that of molecular functions and cell components (Figure 3)”. Table 2 does not present overrepresentation tests. It is poorly informative.

Thanks for your suggestions. We had moved the original Figure 3 to the supplementally material, and the sentence “In addition, we noted that the proportion of classified items of biological processes was greater than that of molecular functions and cell components (Figure 3)” had been removed.

  1. Figure 4 – please reconsider figure title

Thanks for your advices. For your understand, we had added the details legends in Figure 4

  1. Table 3. I understand that you are presetting p-value, not adjusted p or FDR? No significant values? Please clarify.

Thank you very much. I’m very sorry that the difference is not significant if it is adjusted p or FDR. However, our aim is to present the KEGG pathway related to muscle growth and development.

  1. Section 4.5. - in methods section, reference genome and annotation file accession number plus database used is enough. Please provide instead of whole link.

Thanks for your suggestions. We had deleted the whole link.

  1. Section 4.5. - FPKM is not recommended for further normalization with Deseq2 (I understand you used Deseq2 not old Deseq?). You should use raw counts for differential expression analysis. Please reconsider.

Thank you for your suggestion. We will correct it in the future data analysis.

  1. Raw sequencing data as well as processed data e.g. raw counts should be submitted to the public database such as SRA and/or GEO.

Thank you for your suggestion. Raw sequencing data had been submitted to the public database.

  1. It is not clear how many replicates were prepared per condition. If less than 3, I would be forced to change my recommendation on the manuscript.

Thank you for your guidance. We originally set 4 duplicates, but because the error was too large, we removed one

Once again, thank you very much for your guide and comments

Best Regards,

Sincerely yours

Xuemei Yin

College of Marine and Bioengineering, YanCheng Institute of Technology, Yancheng, 224100, Jiangsu, China

E-mail: xmyin1990@163.com

Round 2

Reviewer 1 Report

Now the manuscript is improved and consider for publication.

Minor editing of English language required